# Characterization of Putative Sporulation and Germination Genes in *Clostridium perfringens* Food-Poisoning Strain SM101

**DOI:** 10.3390/microorganisms10081481

**Published:** 2022-07-22

**Authors:** Prabhat K. Talukdar, Mahfuzur R. Sarker

**Affiliations:** 1Department of Biomedical Sciences, College of Veterinary Medicine, Oregon State University, Corvallis, OR 97331, USA; 2Department of Microbiology, College of Science, Oregon State University, Corvallis, OR 97331, USA

**Keywords:** spore formers, spores, spore germination, gene homology

## Abstract

Bacterial sporulation and spore germination are two intriguing processes that involve the expression of many genes coherently. Phylogenetic analyses revealed gene conservation among spore-forming Firmicutes, especially in Bacilli and Clostridia. In this study, by homology search, we found *Bacillus subtilis* sporulation gene homologs of *bkdR*, *ylmC*, *ylxY*, *ylzA*, *ytaF*, *ytxC*, *yyaC1*, and *yyaC2* in *Clostridium perfringenes* food-poisoning Type F strain SM101. The β-glucuronidase reporter assay revealed that promoters of six out of eight tested genes (i.e., *bkdR*, *ylmC*, *ytaF*, *ytxC*, *yyaC1*, and *yyaC2*) were expressed only during sporulation, but not vegetative growth, suggesting that these genes are sporulation-specific. Gene knock-out studies demonstrated that *C. perfringens* Δ*bkdR*, Δ*ylmC*, Δ*ytxC*, and Δ*yyaC1* mutant strains produced a significantly lower number of spores compared to the wild-type strain. When the spores of these six mutant strains were examined for their germination abilities in presence of known germinants, an almost wild-type level germination was observed with spores of Δ*ytaF* or Δ*yyaC1* mutants; and a slightly lower level with spores of Δ*bkdR* or Δ*ylmC* mutants. In contrast, almost no germination was observed with spores of Δ*ytxC* or Δ*yyaC2* mutants. Consistent with germination defects, Δ*ytxC* or Δ*yyaC2* spores were also defective in spore outgrowth and colony formation. The germination, outgrowth, and colony formation defects of Δ*ytxC* or Δ*yyaC2* spores were restored when Δ*ytxC* or Δ*yyaC2* mutant was complemented with wild-type *ytxC* or *yyaC2*, respectively. Collectively, our current study identified new sporulation and germination genes in *C. perfringens*.

## 1. Introduction

Sporulation is a unique developmental process in certain Gram-positive bacteria (Firmicutes) wherein a metabolically inactive cell type (the spore) is formed inside another cell (the mother cell) and ultimately released as a free spore into the environment upon the lysis of the mother cell [1,2,3]. Spores are highly resistant to various physical and chemical stresses that can rapidly destroy the vegetative form of the bacterium [4,5,6]. Certain spore-forming foodborne pathogens utilize some of these resistance properties to maintain their viability in different food settings [7]. Spores are widely available in the environment including the gastrointestinal (GI) tract of humans and animals, contaminated food, water, soil, and carcasses. Spores remain metabolically inactive until external stimuli from the surrounding environment trigger germination, by which the dormant spore structures rapidly transform into metabolically active vegetative form, and as a result, spores trade-off their resistance properties [8]. Spores can also act as infectious agents, as many pathogens produce or release toxins during sporulation or spore germination [9]. To understand the survival and pathogenicity of spore-forming foodborne pathogens, studies are warranted to identify uncharacterized genes that are likely to be involved in the process of spore formation and germination.

*Clostridium perfringens* is a Gram-positive, anaerobic, spore-forming, rod-shaped bacterium and an important member of the Firmicute phylum due to its disease-causing ability including foodborne and non-foodborne GI illnesses and tissue necrosis in humans and animals [10]. In the United States, *C. perfringens* ranks as the second most common cause of bacterial foodborne illnesses, causing one million cases each year and resulting in ~400 million dollars of annual economic loss [11,12]. The disease etiology of *C. perfringens* is mainly due to its ability to produce an array of toxins; more than 20 toxins have been identified so far [13]. Based on the production of six major toxins; alpha-, beta-, iota-, epsilon-, CPE- and NetB-toxins, *C. perfringens* strains are classified as seven toxino-types (Type A-G) [14]. *C. perfringens* enterotoxin (CPE), produced by *C. perfringens* type F strains, is responsible for foodborne and non-foodborne GI illnesses, including 5–10% of all cases of antibiotic-associated diarrhea in humans [15]. The production and release of toxins in *C. perfringens* are often directly regulated by either sporulation or germination machinery. For example, the CPE-encoding gene *cpe* is highly expressed during the early stage of sporulation and the release of CPE happens upon the mother cell lysis in the late stage of the sporulation [16]. *C. perfringens* TpeL toxin is also produced during sporulation [17]. In contrast, alpha toxin, a major *C. perfringens* toxin is produced during vegetative growth [9].

Sporulation has been extensively studied in Bacilli, especially in the model organism *Bacillus subtilis*. Although the general pathway of sporulation is similar among all spore-forming organisms, striking differences in the regulation of sporulation have been observed between Bacilli and Clostridia. In general, sporulation initiates when Spo0A, a key transcriptional regulator receives signals from environmental cues and is activated by phosphorylation through a multicomponent signal transduction pathway, also known as phosphorelay, a well-defined system in *B. subtilis* [18,19]. In Clostridia, no such phosphorelay system is present. In *B. subtilis*, when Spo0A~P levels reach a threshold, it regulates four cell-specific sigma (σ) factors (σ^F^, σ^E^, σ^G,^ and σ^K^) that are confined to the forespore (σ^F^ and σ^G^) or the mother cell (σ^E^ and σ^K^) [2,19]. These σ-factors regulate the expression of genes whose products are involved in the spore-structure and -maturation [20]. As the development of spores proceeds, the forespore is completely engulfed by the mother cell to create a cell within a cell. The inner cell becomes the dormant spore and is released from the mother cell by the lysis [2,19]. Upon release, spores remain dormant in the environment until sensing a favorable environment and resume their metabolic activity through the germination [8].

Although the exact mechanism of spore germination differs between Bacilli and Clostridia, the key events are similar in all spore-forming bacteria. Spore germination initiates when spores sense small molecules termed germinants by a set of inner membrane proteins termed germinant receptors (GRs) [1,8]. The signal of germinants to GRs triggers a series of irreversible biophysical events that leads to the release of monovalent cations (H^+^, Na^+^, and K^+^), and the spore’s core large depot of Ca-DPA, degradation of peptidoglycan-rich cortex layer, core hydration, and small acid-soluble protein (SASP) degradation [1,8]. Two major cortex hydrolysis machinery is present among spore-forming bacteria; SleB/CwlJ system is present in Bacilli and some members of Clostridia and the SleC/SleM system is found in only Clostridia [1].

Both sporulation and germination are complex processes and require the coherent expression of many genes. Genome mining and phylogenetic analyses revealed that many of these genes are highly conserved among the spore-forming bacteria. Bacilli and Clostridia are the major spore-forming Firmicutes and have a distant relationship. Bacilli (aerobic organisms) and Clostridia (anaerobic organisms) share a common ancestor that predates the initial rise in oxygen in the atmosphere approximately 2.3 billion years ago [21] and has orthologs of several genes, especially those involved in the sporulation and germination processes [22]. In a genome-wide phylogenetic study, Galperin et al. classified the conserved genes into three categories: (1) genes present in all spore-forming Bacilli and Clostridia, (2) genes present in all Bacilli and most Clostridia, and (3) genes present in most Bacilli and some Clostridia [22]. Following this study, other researchers characterized some of the signature genes in *B. subtilis*. In one study, Abecasis and colleagues identified three new sporulation genes including *ytaF*, *ylmC*, and *ylzA* [23]. In a separate study, Tragg and colleagues used phylogenetic profiling with reverse-genetics and gene-regulatory studies and identified eight new sporulation genes including *bkdR*, *ylmC*, *ymxH*, *ylxY*, *ylzA*, *ymfB*, *yteA*, and *ylyA* [24]. The authors characterized the expression and function of these genes in sporulation conditions. Another study utilized genome-wide transcriptional analyses to identify signature sporulation genes in *C. perfringens* expressed during sporulation conditions [25].

Despite the importance of *C. perfringens* as human and animal pathogens, knowledge about the molecular mechanisms of *C. perfringens* sporulation and germination remains poorly understood. This is because many uncharacterized genes in the *C. perfringens* genome may have a role in spore formation and germination. Consequently, in our current study, a total of eight putative sporulation gene homologs (i.e., *bkdR*, *ylmC*, *ylxY*, *ylzA*, *ytaF*, *ytxC*, *yyaC1*, and *yyaC2*) were selected to investigate their role in *C. perfringens* sporulation and/or spore germination. Our study demonstrated the following findings: (1) all selected sporulation gene homologs were identified in the *C. perfringens* strain SM101 genome; (2) promoter expression analyses showed that six out of eight genes are sporulation-specific genes, and finally (3) gene knock-out studies demonstrated that four genes (*bkdR*, *ylmC*, *ytxC*, and *yyaC1*) are involved in spore formation, two (*ytxC* and *yyaC2*) are in spore germination, and interestingly one (*ytxC*) in both.

## 2. Materials and Methods

### 2.1. Bacterial Strains, Media, and Culture Conditions

All bacterial isolates used in this study are listed in Appendix A. *C. perfringens* strain SM101, a transformable derivative of type F human food-poisoning strain NCTC8798 [26] was used in this study as the major wild-type strain. All mutant isolates were constructed in the background of the SM101 strain. *Escherichia coli* DH5α was used for cloning purposes. Liquid media used for culturing *C. perfringens* included FTG medium (Fluid thioglycollate medium; Becton Dickinson, Sparks, MD, USA), TGY medium (3% trypticase soy, 2% glucose, 1% yeast extract, 0.1% L-cysteine) and Duncan-Strong (DS) sporulation medium [27]. Brain heart infusion (BHI) agar (Becton Dickinson) was used for culturing *C. perfringens* on a solid medium. For *E. coli* culture, Trypticase soy agar (TSA) (Becton Dickinson) and Trypticase soy broth (TSB) were used as solid and liquid media, respectively. *C. perfringens* cultures on a solid medium were incubated anaerobically using the GasPak^TM^ EZ anaerobe container system with an indicator (Becton Dickinson) at 37 °C. *C. perfringens* cultures in liquid media were incubated in a water bath without shaking at 37 °C.

### 2.2. Construction of gusA-Fusion Plasmids and β-Glucuronidase (GUS) Assay

The expression of putative sporulation genes of *C. perfringens* was examined by fusing DNA upstream of each gene to *E. coli*-*gusA* in pMRS127, an *E. coli*-*C. perfringens* shuttle vector [28]. Briefly, 300 to 500 bp DNA fragments upstream of *bkdR*, *ylmC*, *ylxY*, *ylzA*, *ytaF*, *ytxC*, *yyaC1*, and *yyaC2* from *C. perfringens* SM101, that most likely contain target genes’ promoters, were PCR amplified using primer pairs specific for each gene. All primers used in this study are listed in Appendix A. The forward and reverse primers had SalI and PstI cleavage sites, respectively. The PCR fragments were first cloned into a pCR-XL-TOPO cloning vector (Invitrogen, Carlsbad, CA, USA), to yield the initial plasmids. The SalI-PstI fragments from pCR-XL-TOPO clones, that carry putative promoter fragments of each of *bkdR*, *ylmC*, *ylxY*, *ylzA*, *ytaF*, *ytxC*, *yyaC1*, and *yyaC2*, were cloned between SalI and PstI sites in pMRS127 to create *P_bkdR_-*, *P_ylmC_-*, *P_ylxY_-*, *P_ylzA_-*, *P_ytaF_-*, *P_ytxC_-*, *P_yyaC1_-*, and *P_yyaC2_-gusA* fusions, in the respective GUS plasmids. All plasmids used in this study are listed in Appendix A. These GUS plasmids were introduced by electroporation into *C. perfringens* SM101, and erythromycin (Em)-resistant (Em^R^) (50 μg/mL) transformants were selected. Transformants carrying the GUS plasmids were grown in TGY vegetative medium and DS sporulation medium and assayed for GUS activity as described previously [28]. The GUS-specific activity was expressed in Miller units that were calculated as described previously [28]. GUS plasmids that contain *P_bkdR_-*, *P_ylmC_-*, *P_ylxY_-*, *P_ylzA_-*, *P_ytaF_-*, *P_ytxC_-*, *P_yyaC1_-*, and *P_yyaC2_-gusA* fusions were also electroporated into sporulation negative *C. perfringens* mutant strains; IH101 (*spo0A* mutant) and NM101 (σ^E^ mutant), and the transformants were selected on BHI agar plate containing 20 μg/mL chloramphenicol (Cm) and 50 μg/mL Em. All selected transformants were grown in DS medium and the GUS-specific activity was measured as described above.

### 2.3. Construction of C. perfringens Mutant Strains

To construct the isogenic *C. perfringens* SM101 mutant strain with specific gene inactivation, the ClosTron technique, a group II intron-based TargeTron system [29] modified for *Clostridium* genetics was used in this study. The coding sequences of *bkdR*, *ylmC*, *ytaF*, *ytxC*, *yyaC1* and *yyaC2* genes were first entered into the ClosTron website (https://www.clostron.com) for the generation of primer sets. Each primer set consists of three separate primers (IBS, EBS1d, and EBS2) and one universal primer (EBS universal) (Appendix A). Each of the primer sets was used to generate the retargeted 350 bp intron product by using the TargeTron gene knockout system (Sigma-Aldrich Corporation, St. Louis, MO, USA). The PCR amplified 350 bp intron product was cloned into a pCR-XL-TOPO vector and the re-targeted intron sequence was confirmed by Sanger sequencing at the sequencing facility of the Center for Genome Research and Biocomputing (CGRB), Oregon State University. The HindIII/Bsp14071 fragment from each recombinant pCR-XL-TOPO clone was then cloned between the HindIII and Bsp14071 sites of the pJIR3566 vector [30], generating each gene-specific mutator plasmid (Appendix A). Each mutator plasmid was then electroporated into *C. perfringens* wild-type strain SM101 and the transformants were selected onto BHI agar plates supplemented with 20 μg/mL Cm. The (Cm)-resistant (Cm^R^) transformants were grown twice in TGY broth supplemented with 20 μg/mL Cm followed by five passages in TGY without selection to cure the vector plasmid. The transformants were plated onto BHI agar plates supplemented with 50 μg/mL Em and Em^R^ colonies were streaked onto BHI agar plates containing either 20 μg/mL Cm or 50 μg/mL Em. The colonies that appeared as Em^R^ but Cm-sensitive (Cm^S^) were selected and confirmed for the insertion of the retargeted intron by PCR using the gene-specific detection primers (data not shown). With this approach, we created the following six isogenic *C. perfringens* SM101 mutant strains: Δ*bkdR*, Δ*ylmC*, Δ*ytaF*, Δ*ytxC*, Δ*yyaC1*, and Δ*yyaC2.*

### 2.4. Construction of C. perfringens Complemented Strains

To construct a *ytxC* or *yyaC2* complemented plasmid, a DNA fragment, carrying the promoter sequences and the coding region of the specific gene, was PCR-amplified with Phusion High Fidelity DNA polymerase (Thermo Scientific, Vilnius, Lithuania) using the gene-specific primer pairs (Appendix A) (forward and reverse primers had KpnI and SalI sites, respectively, at their 5′ ends). The amplified PCR fragments were digested with KpnI-SalI and cloned into KpnI-SalI sites of *C. perfringens*/*E. coli* shuttle vector, pJIR750 [31], giving the recombinant plasmids carrying wild-type genes (Appendix A). These complementing plasmids were introduced into the *C. perfringens* Δ*ytxC* and Δ*yyaC2* mutant strains by electroporation and Em^R^ Cm^R^ transformants were selected. The presence of complementing plasmid in each mutant strain was confirmed by PCR analyses.

### 2.5. Growth Kinetics of C. perfringens Strains in Vegetative and Sporulation Medium

To analyze the vegetative culture growth, a 0.2 mL aliquot of overnight FTG cultures of the *C. perfringens* SM101 and mutant derivatives were inoculated into 10 mL of TGY medium and incubated for 3 h at 37 °C. The 3 h TGY culture was inoculated into another set of freshly prepared TGY medium and adjusted to OD_600_ of 0.1. The cultures were incubated at 37 °C and 1 mL aliquot was collected for every 1 h (up to 8 h) and OD_600_ was measured. For the evaluation of *C. perfringens* growth in sporulation conditions, an aliquot of overnight FTG cultures of the *C. perfringens* isolates was inoculated into 10 mL of freshly prepared DS medium and adjusted to OD_600_ of 0.1. The cultures were incubated at 37 °C and 1 mL aliquot was collected for every 1 h (up to 8 h) and OD_600_ was measured.

### 2.6. Sporulation Efficiency

An aliquot (0.4 mL) of overnight FTG cultures of *C. perfringens* SM101 and its mutant derivatives were inoculated into 10 mL of DS medium and incubated at 37 °C overnight. A 10 μL aliquot of DS culture was placed onto a cell counting chamber (Hausser Scientific, Horsham, PA, USA) and the number of spores was counted by observing under the phase-contrast microscope (Leica microsystems, Wetzlar, Germany). For each DS culture, two counting slides were prepared, and the average numbers were recorded. In case of very high or very low/no spore count, we either diluted or concentrated the original spore cultures, respectively. Ten microliters of the concentrated or diluted spore cultures were used to count the spores.

### 2.7. Purification of C. perfringens Spores and Spore Germination Assays

Spores of various *C. perfringens* strains were prepared as described previously [32]. Briefly, *C. perfringens* cultures were inoculated into FTG broth and incubated overnight at 37 °C. To prepare spores, a 0.4 mL of FTG starter culture was inoculated into 10 mL of DS medium and incubated for 18–24 h at 37 °C. The presence of spores was confirmed by the phase-contrast microscopy and similar methods were repeated until >70% spores were observed in DS culture. The high volume of sporulating cultures was prepared by scaling up this procedure. Finally, spores were purified by repeated washing with ice-cold sterile distilled water followed by gradient centrifugation using 56% Nycodenz (Accurate Chemical & Scientific Corp., Westbury, NY, USA), until spore suspensions were >99% free of cell debris, sporulating cells, and germinating cells. The purified spores were adjusted to OD_600_ of 6.0 with sterile distilled water and stored at −20 °C until used.

Spore germination assay was conducted as described previously [33]. Briefly, purified spores (OD_600_ of 6.0) of each *C. perfringens* isolate were subjected to heat activation at 80 °C for 10 min, cooled down at room temperature for 5 min, and pre-incubated at 37 °C for 10 min. Then, heat-activated spores of OD_600_ of 1 were incubated with 100 mM of each of KCl or L-cysteine in 25 mM Tris-HCl buffer pH (7.0) at 37 °C for 90 min in a total volume of 0.2 mL in a 96-well microtiter plate. Spore germination was routinely monitored by measuring OD_600_ of spores-germinant solution at 10 min intervals for up to 90 min by using the Smartspec 3000 spectrophotometer (BioTek Instrument Inc., Winooski, VT, USA). Spore germination was also observed by phase-contrast microscopy after 90 min post-incubation, where the fully germinated spores changed from phase-bright to phase-dark. The extent of spore germination was calculated by measuring the decrease in OD_600_ and was expressed as a percentage of initial OD_600_. All values reported are averages of three separate experiments performed on at least three independent spore preparations, and individual values varied by less than 10% from the average values shown.

### 2.8. Measurement of DPA Release by C. perfringens Spores

DPA release from spores during nutrient and non-nutrient germination was measured by heat activating (80 °C, 10 min) a spore suspension (OD_600_ of 1.5), cooling, and incubating with pre-heated germinants at 37 °C for 90 min. A 1 mL aliquot was centrifuged in a microcentrifuge (13,200 rpm, 2 min), and the spore pellet was washed four times with 1 mL of distilled water. Control experiments were completed for each experiment and reveal that losses of spores due to these multiple centrifugations were <10% of the initial amount, and appropriate corrections for such losses were made accordingly. The residual spore DPA content was determined by boiling the samples for 60 min, cooling them on ice for 5 min, centrifugation at 13,200 rpm in a microcentrifuge for 5 min, and measuring the OD_270_ of the supernatant fluid as described previously [34]. The DPA content of the initial dormant spores was measured by boiling an aliquot (1 mL) for 60 min, centrifugation at 13,200 rpm in a microcentrifuge for 5 min, and measuring the OD_270_ of the supernatant fluid as described previously [34].

### 2.9. Spore Outgrowth and Colony Forming Efficiency

Spore outgrowth of *C. perfringens* isolates was determined as described previously [35]. Briefly, heat-activated spores of *C. perfringens* isolates were inoculated into pre-warmed 10 mL TGY broth to a final OD_600_ of 0.1 and incubated anaerobically at 37 °C. The OD_600_ of the culture was measured at various time points (every 30 min for up to 4 h). The colony-forming efficiency was determined with both untreated and decoated spores. Purified spores were decoated as described previously [34]. Briefly, cell pellets were collected from 1 mL of 8 h DS sporulating cultures and chemically treated any spores present in DS with 1 mL 50 mM Tris-HCl (pH 8.0), 8 M urea, 1% (*w*/*v*) sodium dodecyl sulfate (SDS) and 50 mM dithiothreitol (DTT) for 90 in at 37 °C. A similar decoating treatment was performed with the purified spore suspensions at OD_600_ of 1.0, and the remaining spores were washed three times with 150 mM NaCl and twice with water. Both untreated and decoated spores (OD_600_ 1.0) were heat-activated (80 °C for 10 min) and then aliquots of various dilutions were plated onto BHI agar with or without lysozyme (1 μg/mL). The plates were incubated at 37 °C anaerobically for 24 h, and the colonies were counted.

### 2.10. Statistical Analyses

Statistical analyses were performed with GraphPad Prism 6.0 h (La Jolla, CA, USA). The specific statistical tests are indicated in the figure legends.

## 3. Results

### 3.1. Identification of Putative Sporulation Genes in C. perfringens SM101

To identify a list of new sporulation genes in *C. perfringens* that were previously uncharacterized, a search for homologs of sporulation proteins in *B. subtilis* was conducted. The complete reference genome sequences of *C. perfringens* type F strain SM101 (GenBank: CP000312.1) and *Bacillus subtilis* subsp. *subtilis* str. 168 (GenBank: NC_000964.3) from the NCBI genome database was used for this purpose. To minimize the list of target genes for this study, two previously published reports that identified and characterized new sporulation genes in *B. subtilis* were considered [23,24]. By using the blastp algorithm, the protein homolog encoded by *bkdR*, *ylmC*, *ylxY*, *ylzA*, *ymxH*, or *ytaF* was searched in the *C. perfringens* SM101 genome. The protein products of six genes have been found in the *C. perfringens* genome with varying degrees of homology (23–80% amino acid identity) (Table 1). However, CPR1731 in *C. perfringens* was found as the homolog for both YmxH and YlmC in *B. subtilis*, and we designated this protein as YlmC. Three additional genes *CPR1859*, *CPR2154*, and *CPR2663* encoding YtxC, YyaC1, and YyaC2, respectively, were selected as all of these gene products have been listed as ‘putative sporulation protein’ in the reference genome database. Together, a total of eight genes (i.e., *bkdR*, *ylmC*, *ylxY*, *ylzA*, *ytaF*, *ytxC*, *yyaC1*, and *yyaC2*) were selected to investigate their role in *C. perfringens* sporulation and/or spore germination.

Next, we analyzed the gene organization and position of each gene in the *C. perfringens* strain SM101 reference genome (GenBank: CP000312.1). The transcriptional analyses showed that all genes except *ylzA* are expressed individually and are not in the part of an operon (Figure 1). *ylzA* is expressed as the first gene of a bicistronic operon. The protein domains were also analyzed from the UniProtKB database for all proteins encoded by the eight genes.

The domains of eight proteins are shown in Figure 2. Secondary protein structure analyses were conducted to observe the presence of the transmembrane (TM) domain in these proteins. TMs are membrane-spanning domains and are part of the GRs, integral membrane proteins in spore-forming bacteria. The TM analysis using the TMHMM Server v.2.0. predicted one TM in YlxY and six TMs in YtaF. By SignalP 5.0, we also analyzed the signal peptide sequence of all eight proteins. Two of eight proteins have signal peptide sequences, suggesting that these proteins may be processed after maturation.

### 3.2. Expression of Putative Sporulation Genes

To evaluate the expression of selected putative sporulation genes in *C. perfringens*, the promoter region of each of the genes was fused with *E. coli*-*gusA* and the GUS-specific activity was measured after introducing the *gusA*-fusions into *C. perfringens* SM101 as described in Material and Methods. The GUS-specific activity was increased for the promoter of *ylmC* (~700-fold), *bkdR* (~150-fold), *yyaC1* (~100-fold), *ytaF* (~180-fold), *ytxC* (~130-fold), and *yyaC2* (~20-fold) during sporulation (Figure 3). However, under similar experimental conditions, very little or no increase in GUS-specific activity was observed for the promoter of *ylxY* or *ylzA*. While GUS-specific activity from *bkdR-*, *ylmC-*, *ytaF*- or *ytxC*-promoter-*gusA* fusion was increased within 2–4 h of incubation, the GUS activity from *yyaC1*-or *yyaC2-gusA* was increased after 8–10 h. These results suggest that *bkdR*, *ylmC*, *ytaF*, and *ytxC* are expressed during the early stage of sporulation, whereas *yyaC1* and *yaaC2* during the late stage of sporulation. No GUS-specific activity was observed with any of the promoter-*gusA* fusion during vegetative growth (data not shown), suggesting that *bkdR*, *ylmC*, *ytaF*, *ytxC*, *yyaC1*, and *yyaC2* are sporulation-specific genes. To confirm this, *gusA*-fusion plasmids were incorporated into *C. perfringens* strains IH101 (lacking Spo0A, a master regulator of sporulation) and NM101 (lacking sporulation-specific σ-factor, σ^E^) and GUS-specific activities were measured. No significant GUS-specific activity was observed with any of the promoter-*gusA* fusion in IH101 or NM101 background compared to wild-type SM101 (Figure 3). Collectively, our results confirmed that *bkdR*, *ylmC*, *ytaF*, *ytxC*, *yyaC1* and *yyaC2* are sporulation-regulated genes and are dependent on expressions of *spo0A* and *sigE*.

### 3.3. Growth and Spore Formation by C. perfringens Mutants

To evaluate the role of the newly identified sporulation gene homologs (*bkdR*, *ylmC*, *ytaF*, *ytxC*, *yyaC1*, and *yyaC2*), we introduced the null mutation in each of the genes and compared sporulation and/or germination phenotypes of the mutants with that of wild-type. When mutant and wild-type strains were grown in vegetative and sporulation conditions for 12 h at 37 °C, no significant difference in the growth rate among the mutants compared to the wild-type strain was observed (data not shown).

When we compared the spore-forming ability of *C. perfringens* Δ*bkdR*, Δ*ylmC*, Δ*ytaF*, Δ*ytxC*, Δ*yyaC1*, and Δ*yyaC2* mutant with that of wild-type SM101, an almost wild-type level spore formation was observed with Δ*ytaF* and Δ*yyaC2* mutant; and a significant decrease in spore-forming efficiency was observed with Δ*bkdR*, Δ*ylmC*, Δ*ytxC* or Δ*yyaC1* mutant compared to wild-type SM101 (Figure 4), suggesting that *bkdR*, *ylmC*, *ytxC* or *yyaC1* might play a role in spore formation by *C. perfringens* strain SM101.

### 3.4. Germination of Spores of C. perfringens Mutant and Complemented Strains

Next, we compared the germination ability of spores of Δ*bkdR*, Δ*ylmC*, Δ*ytaF*, Δ*ytxC*, Δ*yyaC1*, or Δ*yyaC2* mutant with that of spores of wild-type in presence of known germinants. As expected, wild-type SM101 spores germinated well with KCL, L-cysteine, or L-lysine i.e., a significant OD_600_ decrease (~50–65%) was observed when SM101 spores were incubated with 100 mM of each of these germinants in 25 mM Tris-HCL (pH 7.0) buffer at 37 °C for 90 min (Figure 5). However, under similar experimental conditions, SM101 spores did not germinate with 25 mM Tris-HCL (pH 7.0), as ~10–15% OD_600_ decrease was observed (Figure 5). These results were confirmed by phase-contrast microscopy, ~80–90% of SM101 spores became phase-dark after 90 min of incubation with KCL, L-cysteine, or L-lysine, while ~90% of SM101 spores remained phase bright in the presence of 25 mM Tris-HCL (pH 7.0) buffer (data not shown). When spores of mutant strains were incubated with 100 mM of each of KCL, L-cysteine, or L-lysine in 25 mM Tris-HCL (pH 7.0) at 37 °C for 90 min, almost wild-type level germination was observed with spores of Δ*ytaF* or Δ*yyaC1* mutant; and slightly lower level germination with spores of Δ*bkdR* or Δ*ylmC* mutant (Figure 5). These results suggest that *bkdR*, *ylmC*, *ytaF*, and *yyaC1* play no role in *C. perfringens* spore germination.

In contrast, spores of Δ*ytxC* or Δ*yyaC2* mutant did not germinate with KCl or L-cysteine; a negligible level of OD_600_ decrease was observed when Δ*ytxC*- or Δ*yyaC2*-spores were incubated with 100 mM of germinants (KCl or L-cysteine) in 25 mM Tris-HCL (pH 7.0) (Figure 6A,B). This level of OD_600_ decrease was almost similar to that obtained with Δ*ytxC*- or Δ*yyaC2*-spores incubated in 25 mM Tris-HCl (pH 7.0) for 90 min (data not shown). These results were confirmed by using phase-contrast microscopy, ~ 90% of Δ*ytxC* or Δ*yyaC2* spores remained phase bright after 90 min of incubation with 100 mM of KCl or L-cysteine in 25 mM Tris-HCl (pH 7.0) (data not shown). These results suggest that *ytxC* and *yyaC2* are involved in *C. perfringens* spore germination. The spore germination defect in Δ*ytxC* or Δ*yyaC2* mutant was restored to nearly wild-type level when complemented with the wild-type *ytxC* or *yyaC2* gene, respectively (Figure 6A,B), further confirming that *ytxC* and *yyaC2* are required for full germination of spores of *C. perfringens.*

As the DPA release is another indication of spore germination, we also measured the spore’s DPA content in presence of KCl, L-cysteine for wild type, Δ*ytxC* mutant, Δ*yyaC2* mutant, and complemented strains. From previous studies, it was shown that about 50% of DPA was released in pre-heated *C. perfringens* spores even without the presence of any germinants [36]. In this study, we found that spores of Δ*ytxC* and Δ*yyaC2* mutants released around 50–60% DPA during germination with 100 mM KCl, and 100 mM L-cysteine compared to wild-type, suggesting *ytxC* and *yyaC2* are required for DPA release (Figure 7A,B). Furthermore, the DPA release defect in Δ*ytxC* or Δ*yyaC2* mutant was restored to almost wild-type level by complementing Δ*ytxC* or Δ*yyaC2* mutant with wild-type *ytxC* or *yyaC2*, respectively. Collectively, these results further confirmed that *ytxC* and *yyaC2* are essential for *C. perfringens* SM101 spore germination.

### 3.5. Outgrowth and Colony Formation by Spores of C. perfringens ytxC and yyaC2 Mutants and Complemented Strains

The germination defects observed in Δ*ytxC* and Δ*yyaC2* spores suggested that these spores might have lower outgrowth and colony-forming efficiencies than that of wild-type SM101 spores, as was observed with spores of other *ger* genes mutants [32,35]. In a rich TGY medium, spores of wild-type strain SM101 were able to initiate outgrowth after incubation for 90 min at 37 °C and continue growing until up to 4 h (Figure 8). Under similar experimental conditions, spores of Δ*ytxC* or Δ*yyaC2* mutant were unable to outgrow during 4 h incubation. However, almost wild-type level outgrowth was observed when Δ*ytxC* or Δ*yyaC2* mutant was complemented with wild-type *ytxC* or *yyaC2* gene, respectively, indicating that *ytxC* and *yyaC2* are required for spore outgrowth.

The severity of the outgrowth defect of Δ*ytxC* or Δ*yyaC2* spores suggested that the colony-forming efficiency of Δ*ytxC* or Δ*yyaC2* spores should be lower than that of wild-type spores. This hypothesis was tested by plating heat-activated spores of wild-type, mutant, and complemented strains on BHI agar and incubating them anaerobically for 24 h at 37 °C. The colony-forming efficiency of Δ*ytxC* or Δ*yyaC2* spores was ~10^5^-fold lower than that of wild-type spores (Table 2). However, wild-type level colony-forming efficiency was observed when Δ*ytxC* or Δ*yyaC2* mutant was complemented with *ytxC* or *yyaC2* gene, respectively. Also, the defect of the colony-forming ability of Δ*ytxC* or Δ*yyaC2* spores was eliminated when Δ*ytxC* or Δ*yyaC2* spores were decoated and plated onto BHI agar containing lysozyme, indicating that these mutant spores were viable, but probably incapable of forming colonies due to germination defects. Collectively, our results indicate that the products of the *ytxC* and *yyaC2* are essential for completing germination and outgrowth and thus for colony formation in the BHI medium.

## 4. Discussion

In this study, by homology search, we found *B. subtilis* sporulation gene homologs of *bkdR*, *ylmC*, *ylxY*, *ylzA*, and *ytaF* in *C. perfringenes* food-poisoning Type F strain SM101. We also identified three putative sporulation genes; *ytxC*, *yyaC1*, and *yyaC2* from the genome sequence of strain SM101. We showed that six (*bkdR*, *ylmC*, *ytaF*, *ytxC*, *yyaC1*, and *yyaC2*) of these eight genes were expressed during sporulation, and under the control of key transcriptional regulator Spo0A and early transcription sigma factor, σ^E^. Part of our findings are similar to the previous reports in *B. subtilis*, where the expression of *ylxY*, *ytaF*, and *ylmC* was shown to be sporulation specific and under the control of σ^E^ [23,24]. This study also identified *bkdR*, *ylmC*, *ytxC*, and *yyaC1* are sporulation genes and the proteins encoded by these genes have a role in *C. perfringens* sporulation. Most importantly, our study identified and confirmed *ytxC* and *yyaC2* as two new germination genes in *C. perfringens* strain SM101.

We found that *ylxY* and *ylzA* did not express during *C. perfringens* sporulation. In *B. subtilis*, the *ylxY*-*lacZ* reporter was induced between 1 and 2 h of sporulation, the gene expression is σ^E^-dependent and under the negative control of the mother cell-specific regulator SpoIIID [24]. YlxY is a polysaccharide deacetylase and may have potential catalytic residues that are required to activate cell wall hydrolase CwlJ to efficiently degrade the cortex during spore germination in *B. subtilis* [37]. In another study, Abecasis and colleagues [23] observed the *ylzA* expression during vegetative growth and in the mother cell following asymmetric division at the onset of sporulation suggesting the expression of *ylzA* at the late stage of sporulation and possibly under the control of σ^F^. YlzA has been reported to have a role in the expression of extracellular matrix and biofilm formation in *B. subtilis* [38]. The lack of gene expression of *ylxY* and *ylzA* genes in this study requires further investigation with other *C. perfringens* strains. One possibility is that *ylzA* expression happens during the later stage of sporulation in *C. perfringens* (after 12 h) and could not be detected by the GUS assay in this study due to earlier sampling time points. Despite the lack of expression, it is interesting to see whether *ylxY* and *ylzA* have a role in *C. perfringens* similar to *B. subtilis*.

The inactivation of either *bkdR*, *ylmC*, *ytxC*, or *yyaC1* genes resulted in a significant decrease in spore formation compared to the wild-type strain. In *B. subtilis*, *bkdR* and *ylmC* are identified as sporulation genes. However, experimental data showed a negligent defect in spore formation with *bkdR* and *ylmC* mutants in *B. subtilis* [24]. BkdR encoded by the *bkdR* gene is a σ^54^-dependent DNA-binding transcriptional regulator and found to control the σ^L^-dependent isoleucine and valine degradation pathway in *B. subtilis* [39]. In *B. thuringiensis*, the *bkdR* gene cluster is composed of eight genes and the deletion of *bkdR* decreased the motility of cells, but did not have any effect on growth, sporulation efficiency, and Cry protein production [40]. In *B. subtilis*, *bkdR* is in a seven-gene operon that is involved in the branched-chain amino acid utilization [24]. However, in *C. perfringens*, *bkdR* is a single gene and not a part of an operon. BkdR may act as a transcriptional regulator for other genes associated with sporulation in *C. perfringens*.

YlmC and YmxH are paralogs in *B. subtilis*, and the disruption of either *ylmC* or *ymxH* did not affect sporulation, while disruption of both *ylmC* and *ymxH* impaired the sporulation [23]. This suggests that *ylmC* and *ymxH* may have a redundant role in the mother cell in *B. subtilis*. In another study, Traag and colleagues found that the deletion of *ylmC*, as well as a double mutant of *ylmC* and *ymxH*, caused a slight competitive advantage over a wild-type strain under sporulation-inducing conditions [24]. In *C. perfringens*, there are two YlmC/YmxH family sporulation proteins encoded by *CPR1651* and *CPR1731*. *CPR1651* and *CPR1731* encoded proteins are 91 and 90 amino acids long, respectively, and the amino acid sequence alignment showed there is a 25% sequence similarity between these two proteins. This suggests that these two proteins are paralogs in *C. perfringens* and may have redundant roles. In this study, we identified the *CPR1731* (*ylmC*) as a sporulation gene as the inactivation of this gene decreased the number of spores in *C. perfringens*. Future studies should look at the single deletion of *CPR1651* as well as the double deletion of *CPR1731* and *CPR1651* genes to confirm their role in sporulation. Both proteins have the PRC-barrel domain, an ancient domain that is present in many biological systems including bacteria, archaea, and plants, and have widespread functions ranging from RNA processing to photosynthesis [41].

In this study, we did not find any role of *ytaF* in *C. perfringens* spore formation. In *B. subtilis*, the disruption of *ytaF* caused a reduction in the number of heat-resistant spores [23], although the mechanism of *ytaF* in sporulation is not well understood. However, it has been hypothesized that YtaF is most likely a membrane-associated Ca^2+^-binding protein, and could be involved in the activation of unknown Ca^2+^-dependent proteins required for the sporulation [23]. In addition, YtaF is part of the pathway by which the Ca-DPA accumulates in spores. In *C. perfringens*, YtaF has six predicted TM domains in its secondary structure and indicates its location in either spore’s inner or outer membrane. Interestingly, we did not see any significant decrease in spore germination with *ytaF* mutant. Further studies with YtaF in other *C. perfringens* strains are required.

One of the important findings of this study is to identify *ytxC* and *yyaC2* as new germination genes for *C. perfringens* strain SM101. Most importantly, YtxC exhibited a dual role in sporulation and spore germination as evident from this work. YtxC is a 292 amino acid residue protein, which has the YtxC-like sporulation domain similar to *B. subtilis* YtxC. YtxC also possesses one TM in its secondary structure and a signal peptide sequence. This suggests that it is synthesized inside the cells and upon processing it may transfer to the spore’s inner or outer cell membrane. Further studies are required to identify the localization of YtxC and whether this protein act as GR in *C. perfringens* germination.

In *Clostridium acetobutylicum*, a single YyaC is present and has germination protease (GPR)-like activity and SASP specificity [42]. In contrast, there are two YyaC-like proteins in *C. perfringens*, and our study identifies YyaC1 has a role in sporulation and YyaC2 in spore germination. These two proteins are paralogs and the protein structure contains a single DUF1256 domain similar to YyaC of *C. acetobutylicum*. It is possible that YyaC2 may act as a GPR in *C. perfringens*. However, this study did not confirm the role of YyaC2 as GPR. Further study with *yyaC1*, *yyaC2* double knockout mutant in conjunction with *yyaC1* and *yyaC2* single deletion mutants are required to know their role in *C. perfringens* sporulation and spore germination.

In summary, the current study identified four genes that are involved in sporulation, and two genes that have a role in germination in the *C. perfringens* SM101 strain. The findings of this study are significant as many genes are yet to be identified and characterized to understand *C. perfringens* physiology. This study also exhibits the importance of homology search for similar proteins that have been identified in other bacteria. By identifying and characterizing unknown genes, it would be possible to understand the complex sporulation and germination processes in spore-forming bacteria, and thus facilitate the development of new sporicidal strategies.

## Figures and Tables

**Figure 1 microorganisms-10-01481-f001:**
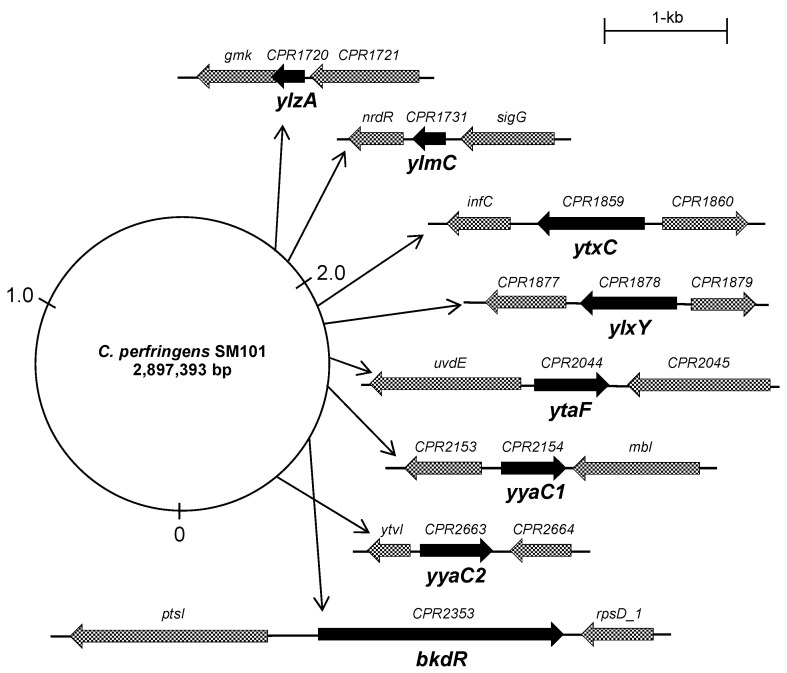
Genomic arrangement of putative sporulation genes in *C. perfringens* strain SM101. Arrangement and position of *ylzA*, *ylmC*, *ytxC*, *ylxY*, *ytaF*, *yyaC1*, *yyaC2*, and *bkdR* genes are shown in the *C. perfringens* SM101 genome. The dark arrows indicate the targeted gene that was selected for this study. The dotted arrows represent the upstream and downstream of targeted genes. Thin arrows represent the position of the genes in the SM101 genome.

**Figure 2 microorganisms-10-01481-f002:**
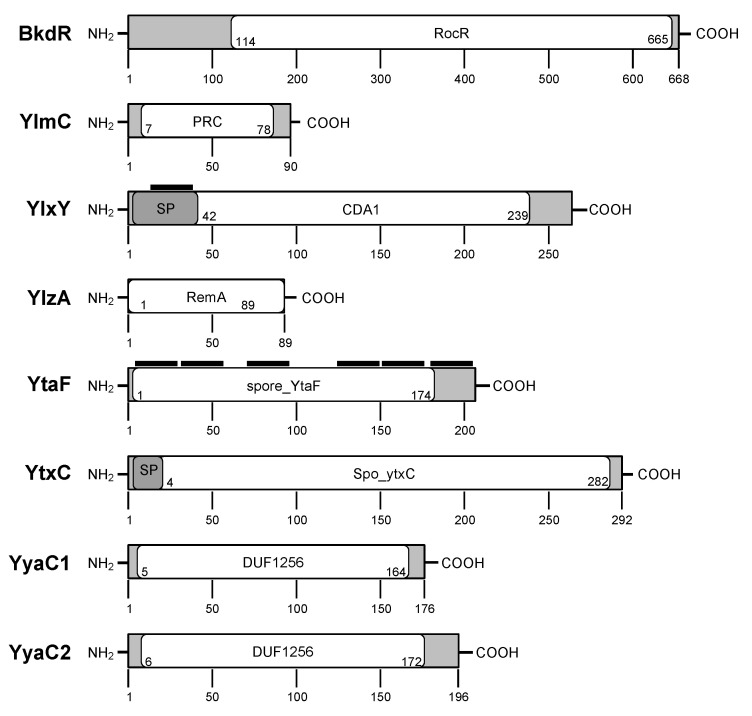
Schematic diagram of protein structure and predicted domains in *C. perfringens* strain SM101. SP denotes signal peptide sequence. The black bar on top of the protein indicates the transmembrane domain.

**Figure 3 microorganisms-10-01481-f003:**
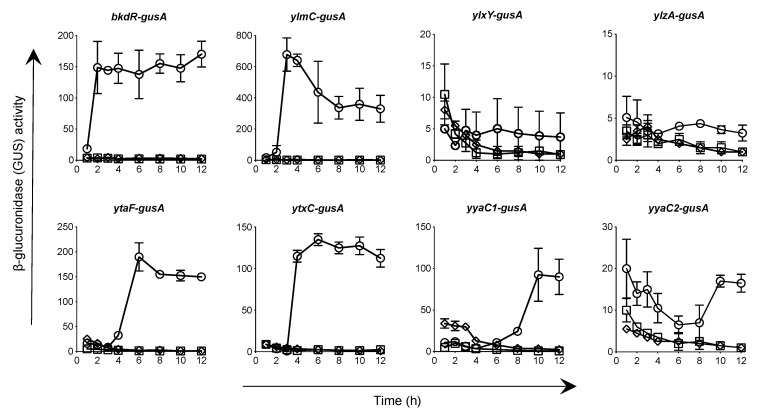
Promoter expression of candidate sporulation genes in *C. perfringens* strains. The *gusA*-promoter fusion constructs of *bkdR*, *ylmC*, *ylxY*, *ylzA*, *ytaF*, *ytxC*, *yyaC1*, and *yyaC2* were introduced into SM101 (wild-type) (○), Δ*spo0A* (□), and Δ*sigE* (◊) mutant strains and β-glucuronidase (GUS) activity was measured as described in Material and Methods. The data represent are the mean of three independent experiments, and time zero denotes the time of inoculation of cells into the DS medium. Note that symbols □ and ◊ are overlapped.

**Figure 4 microorganisms-10-01481-f004:**
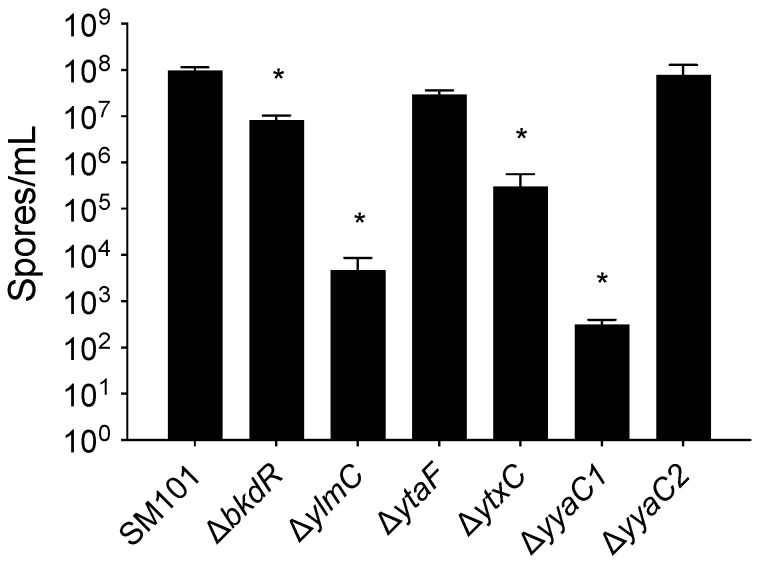
Spore forming efficiency of *C. perfringens* strains. The spore-forming efficiencies of *C. perfringens* wild-type SM101 and its mutant derivatives were measured by the direct microscopic count from 12 h culture in DS sporulation medium as described in Material and Methods. Statistical significance was determined by one-way ANOVA followed by Dunnett’s multiple comparison test. The asterisk (*) denotes the significant differences (*p* < 0.05) in the values of each isolate compared to the value obtained from the wild-type strain.

**Figure 5 microorganisms-10-01481-f005:**
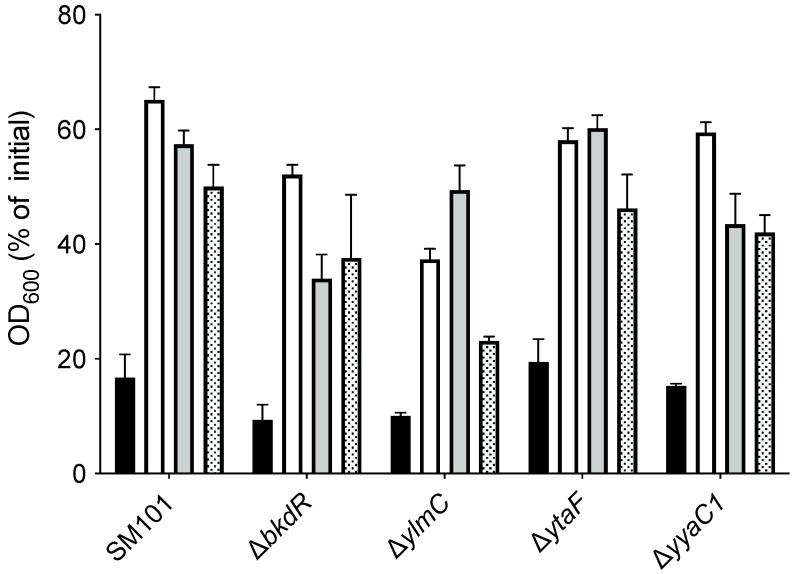
Germination of *C. perfringens* spores with various germinants. Heat-activated spores of *C. perfringens* wild-type strain SM101 and its mutant derivatives were germinated with 25 mM Tris-HCl (pH 7.0) (buffer control) (black bar); 100 mM KCl in 25 mM Tris-HCl (pH 7.0) (white bar); 100 mM L-cysteine in 25 mM Tris-HCl (pH 7.0) (grey bar); and 100 mM L-lysine in 25 mM Tris-HCl (pH 7.0) (dotted bar) at 37 °C for 90 min, and the OD_600_ was measured as described in Material and Methods. The data represents the means of three separate experiments and the standard errors of the mean were less than 25% of the mean.

**Figure 6 microorganisms-10-01481-f006:**
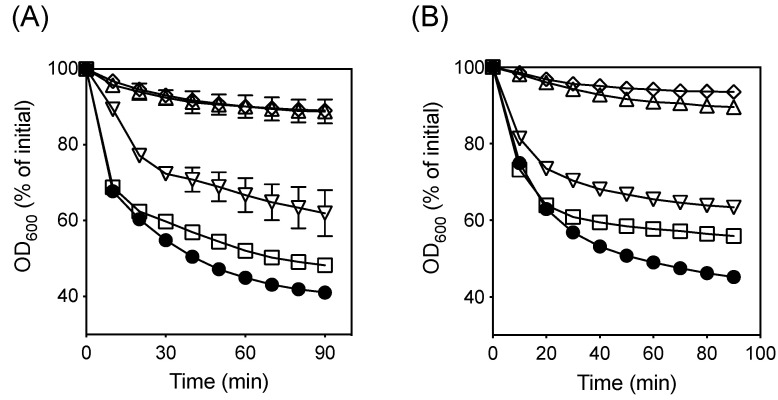
Germination of *C. perfringens* Δ*ytxC* and Δ*yyaC2* mutants and complemented strains with various germinants. Heat-activated spores of *C. perfringens* SM101 (wild-type) (●), Δ*ytxC* mutant (◊), Δ*ytxC+ytxC* (Δ*ytxC* mutant complemented with wild-type *ytxC*) (□), Δ*yyaC2* mutant (Δ), and Δ*yyaC2+yyaC2* (Δ*yyaC2* mutant complemented with wild-type *yyaC2*) (∇), were germinated with (**A**) 100 mM KCl in 25 mM Tris-HCl (pH 7.0) and (**B**) 100 mM L-cysteine in 25 mM Tris-HCl (pH 7.0) at 37 °C for 90 min, and the OD_600_ was measured as described in Material and Methods. The data represents the means of three separate experiments and the standard errors of the mean were less than 25% of the mean. There was no significant decrease in the OD_600_ for all *C. perfringens* spores incubated in buffer alone (data not shown).

**Figure 7 microorganisms-10-01481-f007:**
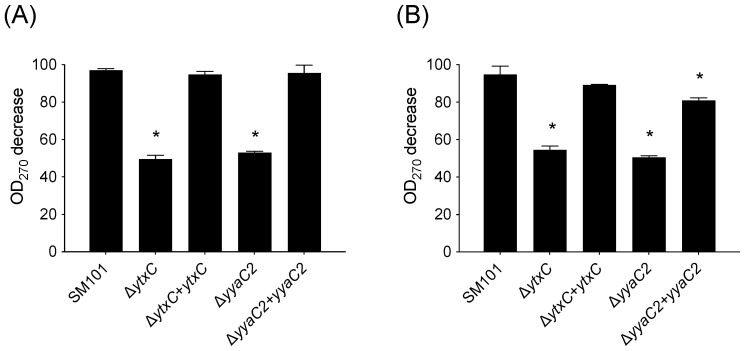
DPA release during germination of spores of *C.*
*perfringens* mutant and complemented strains. Heat-activated spores of *C. perfringens* strain SM101 (wild-type), Δ*ytxC* mutant, Δ*ytxC+ytxC* (Δ*ytxC* mutant complemented with wild-type *ytxC*), Δ*yyaC2* mutant, and Δ*yyaC2+yyaC2* (Δ*yyaC2* mutant complemented with wild-type *yyaC*), were incubated with (**A**) 100 mM KCl in 25 mM Tris-HCl (pH 7.0); and (**B**) L-cysteine in 25 mM Tris-HCl (pH 7.0) at 37 °C for 90 min and DPA release was monitored by measuring OD_270_ as described in Material and Methods. The data represents the means of three replicates and error bars represent standard deviations. Statistical significance was determined by one-way ANOVA followed by Dunnett’s multiple comparison test. The asterisk (*) denotes the significant differences (*p* < 0.05) in the values of each isolate compare to the value obtained from the wild-type strain.

**Figure 8 microorganisms-10-01481-f008:**
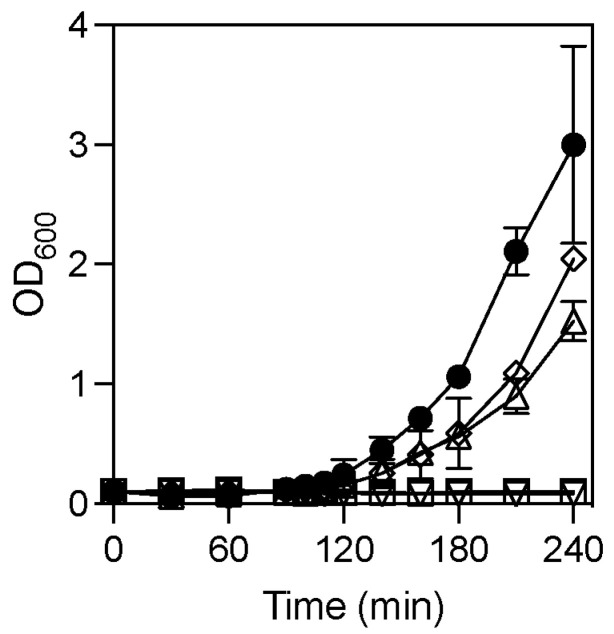
The outgrowth of spores of *C. perfringens* wild-type, mutant, and complemented strains. Heat-activated spores of *C. perfringens* strain SM101 (wild-type) (●), Δ*ytxC* mutant (∇), Δ*ytxC+ytxC* (Δ*ytxC* mutant complemented with wild-type *ytxC*) (◊), Δ*yyaC2* mutant (□), and Δ*yyaC2+yyaC2* (Δ*yyaC2* mutant complemented with wild-type *yyaC*) (Δ) were inoculated in pre-warmed TGY broth and the OD_600_ was measured as described in Material and Methods. The data represents the means of three individual replicates and error bars represent standard deviations.

**Table 1 microorganisms-10-01481-t001:** Putative sporulation genes in *C. perfringens* strain SM101.P.

Gene	Locus Tag ^a^	Predicted Product ^b^	Identity (%) ^c^
*ylzA*	CPR1720	Conserved hypothetical protein	80
*ylmC*	CPR1731	PRC-barrel domain protein	37
*ytxC*	CPR1859	Putative sporulation protein YtxC	23
*ylxY*	CPR1878	Polysaccharide deacetylase family protein	31
*ytaF*	CPR2044	Putative sporulation protein YtaF	34
*yyaC1*	CPR2154	Putative sporulation protein YyaC	35
*bkdR*	CPR2353	σ^54^ dependent transcriptional regulator	35
*yyaC2*	CPR2663	Putative sporulation protein YyaC	45

^a^*C. perfringens* SM101 (GenBank: CP000312.1) locus tags. ^b^ Predicted protein products of corresponding genes from NCBI database (GenBank: CP000312.1). ^c^ Amino acid sequence identity was determined by using the Clustal Omega multiple sequence alignment tool and the protein sequence of *C. perfringens* SM101 (GenBank: CP000312.1) and *B. subtilis* strain 168 (GenBank: NC_000964.3) was used.

**Table 2 microorganisms-10-01481-t002:** Colony formation by spores of *C. perfringens* strains ^a^.

Strains	Spore Titer (CFU/mL/OD_600_) ^b^	
	BHI	BHI+Lysozyme ^c^
SM101 (wild-type)	3.05 × 10^7^	1.52 × 10^8^
Δ*ytxC*	3.70 × 10^2^	1.20 × 10^8^
Δ*yyaC2*	2.25 × 10^2^	8.91 × 10^7^
Δ*ytxC+ytxC*	2.95 × 10^7^	8.50 × 10^7^
Δ*ytxC+yyaC2*	1.78 × 10^7^	9.95 × 10^7^

^a^ Heat-activated spores of various strains were plated onto BHI agar with or without lysozyme (1 μg/mL), and colonies were counted after anaerobic incubation at 37 °C for 24 h as described in Material and Methods. ^b^ Titers are the average number of colony-forming units (CFU/mL/OD_600_) determined in three individual experiments, and the variation was less than 15%. ^c^ Spores were decoated, heat-activated, and plated onto BHI agar containing lysozyme (1 μg/mL), and colonies were counted after incubation anaerobically at 37 °C for 24 h.

## Data Availability

Not applicable.

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

*spo0A* mutant with wild-type *spo0A* from other *Clostridium* species. Appl. Environ. Microbiol..

