# Peer review of "Characterization of Putative Sporulation and Germination Genes in Clostridium perfringens Food-Poisoning Strain SM101"

_microorganisms, 2022, doi:10.3390/microorganisms10081481_

Round 1

Reviewer 1 Report

This is a solid piece of work examining a number of candidate genes/proteins that may have a role in the sporulation and/or germination of Clostridium perfringens. The narrative follows a fairly routine work flow – homology searches, promoter analyses, and then null mutant characterisation via a series of generally appropriate experiments. Overall, the work has been performed to a high standard and provides evidence supporting the identity of four novel (for this species) genes/proteins with roles in sporulation and two with roles in germination. Data interpretation is generally fine although there are a few examples that are a stretch too far and not substantiated (detailed below). The manuscript is easy to follow but there are a significant number of typographical errors. There’s scope also to trim the introduction and discussion, parts of which are repetitive. The following should be addressed:

Line 18 – promoters

Line 21 – lower

Line 26 – mutants

Line 36 – as a free spore

Line 42 – delete ‘the’ before external

Line 44 – transform

Line 54 – delete which

Line 60 – is, not are

Line 72 – delete the; also is activated

Line 77 – regulate

Line 82 – until sensing a favorable environment

Line 90 – and SASP

Line 93 – sporulation, lower case s

Line 111 – conditions (and line 212)

Line 112 – pathogens

Line 119 – gene

Line 219 – delete for before overnight (and elsewhere e.g. 227)

Line 235 – sporulating

Line 239 – cooled down; pre incubated

Line 242 – intervals

Line 254 – why was the DPA assay conducted at 40C (germination 37 elsewhere)

Line 283 – delete The

Line 288 – a search

Table 1 – no need for decimal places, round to nearest integer

Line 358 – predicted to have… (and delete the before YtaF)

Figure 2 – predicted domains

Line 392 – remove ‘growth’ after sporulation (here and elsewhere)

Line 395 – Is sporulation even initiated within 2 hrs of incubation?

Line 397 – the early stage

Line 429 – measured

Line 435 – gene

Line 462 – significance

Line 472 – were incubated

Line 475 – delete when SM101

Figure 4 and Section 3.4 – I’m amazed you can purify and concentrate spores of yyaC1 and ylmC to a sufficient for germination assays based on numbers per ml. Could you comment? Could you even see a single spore under the microscope for strain yyac1 given its concentration of 100 per ml given that 10e6 spores per ml is generally 1 spore in the field of view under x100 optics?

Line 509 – were incubated

Line 517 – complemented

Line 518 – confirming

Line 543 – 545 – loss of 50% DPA upon heat activation is remarkable. Surely these spores already lose a significant amount of refractility before germination assays? If so, it seems unlikely that a 65% OD loss would ever be observed thereafter on the plate reader. Could you comment? Perhaps the temperature of heat activation should be reduced? Maybe the data presented in Figure 7 could be explained by enhanced DPA loss during heat activation of ytxC and yyac2 rather than anything to do with germination (leaky SpoVA channel or similar).

Figure 8 – presenting y axis data as % of original OD seems inappropriate here (4000% etc). Why not just OD values, corrected for any variance at t0?

Line 659 – a role

Line 672 – thuringiensis

Line 676-67 – the BdkR; a different. The fact that bdkR appears to be monocistronic is not sufficient evidence to indicate a different role. I would delete this statement.

Line 693 – predicted to have 6 TM (not found). Which membrane – inner or outer (not cell)? A likely membrane location is not enough evidence to suggest a potential role as a GR!

Line 698 -700 – a dual role. Again, the predicted presence of a putative membrane anchor is nowhere near enough evidence to suggest a role as a GR. The vast majority of spore membrane proteins are not GRs.

Line 706 – a role as a..

Line 704 716 – this entire section should be subject to closer examination using appropriate sequence/bioinformatic/structural tools. It should be simple to ascertain if these proteins are homologues of Gpr, and if so, are the catalytic residues present. Justification of the spore count method is not required.

Line 730 – a role

Line 732 – delete the

Line 734 – this sentence needs to be edited to make sense. …facilitate development of new sporicidal strategies (or similar).

Author Response

Point-by-point response to the reviewers’ comments

Reviewer 1

This is a solid piece of work examining a number of candidate genes/proteins that may have a role in the sporulation and/or germination of Clostridium perfringens. The narrative follows a fairly routine work flow – homology searches, promoter analyses, and then null mutant characterisation via a series of generally appropriate experiments. Overall, the work has been performed to a high standard and provides evidence supporting the identity of four novel (for this species) genes/proteins with roles in sporulation and two with roles in germination. Data interpretation is generally fine although there are a few examples that are a stretch too far and not substantiated (detailed below). The manuscript is easy to follow but there are a significant number of typographical errors. There’s scope also to trim the introduction and discussion, parts of which are repetitive.

Response to Reviewer #1:

Thank you for your constructive comments to improve the quality of the manuscript. We have used Grammarly, an online-based writing assistant tool to correct typographical errors. In addition, we have done some editing in the introduction and discussion section to remove the repetitive information. We deleted lines 642-646.

The following should be addressed:

Line 18 – promoters

Response: Accepted and revised in line 18.

Line 21 – lower

Response: Agreed and revised in line 21.

Line 26 – mutants

Response: Done in line 26.

Line 36 – as a free spore

Response: Agreed and revised in line 36.

Line 42 – delete ‘the’ before external

Response: Done in line 43.

Line 44 – transform

Response: Revised in line 44.

Line 54 – delete which

Response: Done in new line 54.

Line 60 – is, not are

Response: Revised in line 60.

Line 72 – delete the; also is activated

Response: Revised in new line 72.

Line 77 – regulate

Response: Done in line 77.

Line 82 – until sensing a favorable environment

Response: Revised in line 82.

Line 90 – and SASP

Response: Revised in line 91.

Line 93 – sporulation, lower case s

Response: Done in line 94.

Line 111 – conditions (and line 212)

Response: Revised in lines 112 and 213.

Line 112 – pathogens

Response: Revised in line 113.

Line 119 – gene

Response: Done in line 120.

Line 219 – delete for before overnight (and elsewhere e.g. 227)

Response: Revised in lines 220 and 231.

Line 235 – sporulating

Response: Done in line 239.

Line 239 – cooled down; pre incubated

Response: Revised in line 243.

Line 242 – intervals

Response: Revised in line 247.

Line 254 – why was the DPA assay conducted at 40C (germination 37 elsewhere)

Response: We apologize for the mistake. The assay was conducted at 37C similar condition to the germination assay. The change has been made in line 259.

Line 283 – delete The

Response: Revised in line 288.

Line 288 – a search

Response: Done in line 293.

Table 1 – no need for decimal places, round to nearest integer

Response: Done.

Line 358 – predicted to have… (and delete the before YtaF)

Response: Revised in line 363.

Figure 2 – predicted domains

Response: Revised in line 387.

Line 392 – remove ‘growth’ after sporulation (here and elsewhere)

Response: Revised in lines 397 and 403.

Line 395 – Is sporulation even initiated within 2 hrs of incubation?

Response: In C. perfringens, many sporulation/germination-specific genes are expressed during the early stage of sporulation. For example, sleC, sleM promoter began to express ~2 h after induction of sporulation (Paredes-Sabja et al. J Bacteriol. 2009 Apr;191(8):2711-20. doi: 10.1128/JB.01832-08).

Line 397 – the early stage

Response: Done in line 402.

Line 429 – measured

Response: Revised in line 434.

Line 435 – gene

Response: We have changed accordingly in line 440.

Line 462 – significance

Response: Done in line 467.

Line 472 – were incubated

Response: Revised in line 477.

Line 475 – delete when SM101

Response: We have changed accordingly in line 480.

Figure 4 and Section 3.4 – I’m amazed you can purify and concentrate spores of yyaC1 and ylmC to a sufficient for germination assays based on numbers per ml. Could you comment? Could you even see a single spore under the microscope for strain yyac1 given its concentration of 100 per ml given that 10e6 spores per ml is generally 1 spore in the field of view under x100 optics?

Response: Thank you for the comment. We used DS sporulation medium for the sporulation efficiency of C. perfringens wild-type and mutant strains. Based on the number of spores, we either diluted or concentrated the original sporulation culture. If the spore count was too high (uncountable), we diluted the cultures and counted the spores. In contrast, if the spore count was too few or there was no spore, we concentrated the culture to count the spores. In the case of ylmC and yyaC1 mutant strains, which produced a significantly low number of spores, we concentrated the sporulation culture by centrifugation and resuspended it with 1/100th volume of miliQ H2O. For example, from a 5 ml DS sporulation medium, we centrifuged the cultures and resuspended with 50 µl of miliQ H2O. For each spore count, 10 µl of the spore samples were placed on a cell counting chamber and observed under a 100× optical viewfinder with a phase-contrast microscope. The total number of spores in the field was calculated using the following formula:

No. of spores × 100

We have added two sentences in the methods section to clarify this (lines 224-227)

Line 509 – were incubated

Response: Done in line 514.

Line 517 – complemented

Response: Done in line 522.

Line 518 – confirming

Response: Revised in line 523.

Line 543 – 545 – loss of 50% DPA upon heat activation is remarkable. Surely these spores already lose a significant amount of refractility before germination assays? If so, it seems unlikely that a 65% OD loss would ever be observed thereafter on the plate reader. Could you comment? Perhaps the temperature of heat activation should be reduced? Maybe the data presented in Figure 7 could be explained by enhanced DPA loss during heat activation of ytxC and yyac2 rather than anything to do with germination (leaky SpoVA channel or similar).

Response: We found in our previous studies that, C. perfringens spores release ~50% basal level DPA upon heat-activation. However, at least ~ 80-90% DPA release is required to complete spore germination. In Fig. 7, ytxC and yyaC2 mutant spores released only ~50% basal level DPA upon heat-activation followed by incubation with germinant, indicating no germinant-induced DPA release, which is consistent with these mutant spores are defective in germination (Fig. 6). In contrast, SM101 wild-type spores released more than 90% DPA in presence of germinant, indicating a significant germinant-induced DPA release, and thus these spores were fully germinated (Fig. 6). The DPA release defects in mutants could be restored in complemented strains.

Figure 8 – presenting y axis data as % of original OD seems inappropriate here (4000% etc). Why not just OD values, corrected for any variance at t0?

Response: We revised figure 8 accordingly. The new figure shows the OD600 values taken at different time points.

Line 659 – a role

Response: Revised in line 664.

Line 672 – thuringiensis

Response: Revised in line 677.

Line 676-67 – the BdkR; a different. The fact that bdkR appears to be monocistronic is not sufficient evidence to indicate a different role. I would delete this statement.

Response: We have deleted the sentence.

Line 693 – predicted to have 6 TM (not found). Which membrane – inner or outer (not cell)? A likely membrane location is not enough evidence to suggest a potential role as a GR!

Response: We agree with the reviewer’s comment that the presence of predicted TM in a given protein structure does not ensure its function as GR. However, in C. perfringens, all identified GRs are located in the spore’s inner membrane. Further experimental evidence is needed to localize both YtaF and YtxC in C. perfringens. We have edited the sentence and removed the phrase ‘may function as GR’.

Line 698 -700 – a dual role. Again, the predicted presence of a putative membrane anchor is nowhere near enough evidence to suggest a role as a GR. The vast majority of spore membrane proteins are not GRs.

Response: We have revised accordingly in lines 714-716. We agree with the reviewer’s comment (see above response). We have removed the phrase ‘may function as GR’.

Line 706 – a role as a..

Response: Revised in line 719.

Line 704 716 – this entire section should be subject to closer examination using appropriate sequence/bioinformatic/structural tools. It should be simple to ascertain if these proteins are homologues of Gpr, and if so, are the catalytic residues present. Justification of the spore count method is not required.

Response: We agree with the reviewer that without further experimental evidence and bioinformatics analysis, we cannot conclude that YyaC protein function as germination protease in C. perfringens. We have changed the section of this paragraph in lines 719-727. We have mentioned that this study did not confirm the function of YyaC2 as GPR and further studies are required. We have also deleted the paragraph describing the justification of the spore count.

Line 730 – a role

Response: Done in line 729.

Line 732 – delete the

Response: Done in line 731.

Line 734 – this sentence needs to be edited to make sense. …facilitate development of new sporicidal strategies (or similar).

Response: We modified this line 727 sentence as suggested, “By identifying and characterizing unknown genes, it would be possible to understand the complex sporulation and germination processes in spore-forming bacteria and thus facilitate the development of new sporicidal strategies”. See new line # 733-735.

Reviewer 2 Report

Clostridium perfringens is a Gram-positive anaerobic bacillus that is widely distributed in nature and causes serious food poisoning in humans. These diseases not only seriously threaten the health of people and animals but also cause enormous economic damage. Therefore, the topic related to the study of their life cycle, including sporulation and germination, is relevant. The article presents the results of experimental studies, which were conducted carefully, considering all the necessary successive stages.

I have a few comments and recommendations.

1. Authors should check and correct the taxonomy and taxonomic names that they mention in the text of the article. For example, the correct spelling is Firmicutes, not Fermicutes (L. 15, 51, 96). Firmicutes is taxonomically a phylum of bacteria, not a class (L. 96) or family (L. 51). Such inaccuracies should be eliminated throughout the text.

2. I recommend that the authors conduct a thorough revision of the literature to provide correct information about the latest achievements in a given topic to the article. Considering that in the text the authors refer not only to original articles, but also to literature reviews, it would be correct if they updated the information in the main reviews that currently describe the current state of research in this area. For example, the authors refer to the review by Uzal et al., 2014, which apparently mentions 17 toxins characterized from Clostridium perfringens, but a quick search among new publications gave me the review by Wang, 2020, where the author mentions more than 20 toxins, indicating specific original publications on this topic. Perhaps a thorough search for relevant publications would have provided even more correct information.

L. 240–241: It is necessary to correctly check the indications of spores to germinant ratio

Author Response

Point-by-point response to the reviewers’ comments

Reviewer 2

Clostridium perfringens is a Gram-positive anaerobic bacillus that is widely distributed in nature and causes serious food poisoning in humans. These diseases not only seriously threaten the health of people and animals but also cause enormous economic damage. Therefore, the topic related to the study of their life cycle, including sporulation and germination, is relevant. The article presents the results of experimental studies, which were conducted carefully, considering all the necessary successive stages.

Response to Reviewer #2: Thank you for your insightful comments.

I have a few comments and recommendations.

  1. Authors should check and correct the taxonomy and taxonomic names that they mention in the text of the article. For example, the correct spelling is Firmicutes, not Fermicutes (L. 15, 51, 96). Firmicutes is taxonomically a phylum of bacteria, not a class (L. 96) or family (L. 51). Such inaccuracies should be eliminated throughout the text.

Response: We apologize for the typographical error. We also agree that Firmicutes is a phylum. We have made necessary corrections in Lines 15, 51, and 97.

  1. I recommend that the authors conduct a thorough revision of the literature to provide correct information about the latest achievements in a given topic to the article. Considering that in the text the authors refer not only to original articles, but also to literature reviews, it would be correct if they updated the information in the main reviews that currently describe the current state of research in this area. For example, the authors refer to the review by Uzal et al., 2014, which apparently mentions 17 toxins characterized from Clostridium perfringens, but a quick search among new publications gave me the review by Wang, 2020, where the author mentions more than 20 toxins, indicating specific original publications on this topic. Perhaps a thorough search for relevant publications would have provided even more correct information.

Response: Thank you for the valuable suggestion. We revised the literature review and added the latest information regarding the number of toxins in line 56.

  1. 240–241: It is necessary to correctly check the indications of spores to germinant ratio

Response: Thank you for your comment. We revised the germination methods, see lines 244-250.
